# Neural Genetic Search in Discrete Spaces

**Hyeonah Kim** [* 1 2] **Sanghyeok Choi** [* 3] **Jiwoo Son** [4] **Jinkyoo Park** [3 4] **Changhyun Kwon** [3 4]

## Abstract

Effective search methods are crucial for improving the performance of deep generative models at test time. In this paper, we introduce a novel test-time search method, *Neural Genetic Search* (NGS), which incorporates the evolutionary mechanism of genetic algorithms into the generation procedure of deep models. The core idea behind NGS is its crossover, which is defined as parent-conditioned generation using trained generative models. This approach offers a versatile and easy-to-implement search algorithm for deep generative models. We demonstrate the effectiveness and flexibility of NGS through experiments across three distinct domains: routing problems, adversarial prompt generation for language models, and molecular design.

## 1. Introduction

Genetic algorithms (GAs) have demonstrated remarkable performance across a diverse range of tasks with discrete search space, from classic combinatorial optimization problems (Holland, 1992; Omara & Arafa, 2010; Vidal et al., 2012; Nagata & Kobayashi, 2013; Mahmoudinazlou & Kwon, 2024) to more recent applications in molecular design (Morris et al., 1998; Jensen, 2019; Nigam et al., 2021; Kerstjens & De Winter, 2022). By maintaining a population of candidate solutions and iteratively applying evolutionary operators, such as crossover and mutation, GAs can systematically explore vast search spaces.

In this work, we investigate how to incorporate the powerful search capabilities of GAs into deep generative models. Previous efforts that combined GAs with deep learning for better search typically applied GAs after the generation

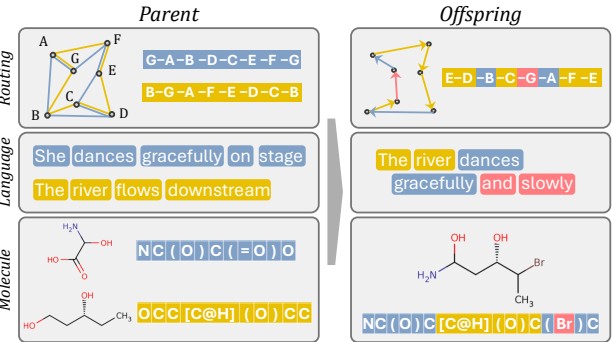

*Figure 1.* Illustrative examples of parents and offspring in various tasks. Offspring are generated through crossover and mutation. Crossover combines two parent chromosomes (Yellow and Blue) by restricting the vocabulary to tokens present in one of the parents. Mutation (Pink) occasionally removes this limitation, promoting the solution diversity.

process to refine the outputs of generative models using existing problem-specific GAs (Ahn et al., 2020; Kim et al., 2024). In contrast, our goal is to integrate the evolutionary principles of GAs directly into the generation process of deep models rather than simply chaining the deep models and GAs.

We present *Neural Genetic Search* (NGS), a GA-inspired new search algorithm for deep generative models. NGS enhances the sequential generation process by incorporating genetic operators, crossover, and mutation. Specifically, we define the crossover as a *parent-conditioned generation*, where the model's vocabulary is restricted to the tokens used in the selected parents, while mutation simply removes this restriction (Figure 1), allowing for more diverse outputs. The generation with the genetic operators proceeds iteratively with an evolving population, steering the generation toward better results over time, similar to traditional GAs.

Since the generation process is still governed by the trained generative models, NGS effectively combine the search capability of GAs with the generative power of deep models. Furthermore, as the proposed genetic operators are problem-agnostic, NGS can be applied to any sequential generative model across various tasks, in contrast to traditional GAs, which require significant effort to design problem-specific operators. NGS also can be easily implemented by adding the population and the parent-conditioned masking rule to

---

[*]Equal contribution [1]Mila - Quebec AI Institute [2]Université de Montréal [3]KAIST [4]Omelet. Correspondence to: Hyeonah Kim <hyeonah.kim@mila.quebec>, Sanghyeok Choi <sanghyeok.choi@kaist.ac.kr>, Changhyun Kwon <chkwon@kaist.ac.kr>.

*Proceedings of the $42^{nd}$ International Conference on Machine Learning*, Vancouver, Canada. PMLR 267, 2025. Copyright 2025 by the author(s).

any existing algorithm with sequential generative models. Finally, compared to single-pass generation, the population-based iterative generation in NGS allows the generation process to adapt over time, improving the quality of generated outputs and even the robustness to distribution shifts.

We evaluated our algorithm in three distinct domains where sequential generative models are widely applied: routing problems, red-teaming language models, and *de novo* molecular design. Our extensive experiments validate that NGS can serve as an effective test-time search method, fulfilling its main purpose. We also found that NGS can replace the conventional decoding algorithms, offering improved robustness. Additionally, NGS can be viewed as an automated genetic algorithm that leverages sequential generative models to learn genetic operators, eliminating the need for extensive labor for algorithm design. These results suggest that NGS has significant potential as a versatile and efficient search strategy, adaptable to a wide range of applications with sequential generative models.

## 2. Background

### 2.1. Sequential Generation in Discrete Space

This paper focuses on the sequential generation of discrete objects to optimize predefined criteria, *i.e.*, maximize reward functions. This setting includes many practical applications, including solution generation for vehicle routing problems, prompt optimization in language models, and atom or fragment-based generation for molecular optimization.

Let $\mathbf{s} \in \mathcal{S}$ represent a sequence that corresponds to a discrete object $\mathbf{x} \in \mathcal{X}$. Without loss of generality and to make the explanation simpler, we assume that the sequence length is $T$, *i.e.*, $\mathbf{s} = (s_1, s_2, \ldots, s_T)$ where $s_t \in V$ is a token from vocabulary $V$. We also denote the partial sequence with length $t - 1$ as $s_{<t}$.

We assume that we have a sequential generative policy $p_\theta$. The probability of a sequence $\mathbf{s}$ being generated by the policy $p_\theta$ is:

$$p_\theta(\mathbf{s}) = \prod_{t=1}^{T} p_\theta(s_t | s_{<t}), \quad (1)$$

where $s_{<1}$ is defined as an empty sequence and $p_\theta(s_t | s_{<t})$ denotes the conditional distribution over next token $s_t$ given the partial sequence $s_{<t}$ modeled by $p_\theta$. In our work, this conditional distribution is given by a pretrained neural network, which is often conditioned on some contexts or constrained with problem-specific constraints.

Note that the mapping from a sequence to a discrete object $g : \mathcal{S} \to \mathcal{X}$ is not necessarily injective, *i.e.*, there can be multiple sequences that correspond to a single object. Thus, the probability of an object $\mathbf{x} \in \mathcal{X}$ being generated is defined as $p_\theta(\mathbf{x}) = \sum_{\mathbf{s}:g(\mathbf{s})=\mathbf{x}} p_\theta(\mathbf{s})$. Additionally, we assume the existence of a reward function $r_\mathcal{X} : \mathcal{X} \to \mathbb{R}$ which evaluates the quality of a generated object $\mathbf{x}$. For instance, in routing problems, we can define $r_\mathcal{X}(\mathbf{x}) = -c(\mathbf{x})$ where $c(\mathbf{x})$ is the length of the route $\mathbf{x}$ that we want to minimize. In molecular optimization, $c$ could represent a property such as binding affinity or drug-likeness, which we aim to maximize. Since this work focuses on sequence generation, we mainly use the reward function over sequences, $r = r_\mathcal{X} \circ g$.

Our task is to maximize the reward function by generating a set of objects using the sequential generative model. Depending on the task, we may also want to maintain the diversity in the generated objects. We provide a more detailed discussion for each problem we considered in Appendix A.

### 2.2. Genetic Algorithm

Genetic algorithms (GAs) are evolutionary-inspired meta-heuristics. By mimicking natural selection and genetic evolution, GAs iteratively refine a population of candidate solutions. The main components of GA are as follows:

- **Chromosome:** A representation of a potential solution, encoded in a suitable format (e.g., a sequence of numbers).

- **Population:** A group of chromosomes that evolve over the iterative refinement by GA.

- **Selection:** Parent chromosomes are selected based on their reward, with chromosomes having higher reward being more likely to be chosen.

- **Crossover:** Parents exchange genetic material to create offspring, combining strengths from both.

- **Mutation:** Random changes to the offspring's genetic material to maintain diversity and explore new regions of the solution space.

- **Replacement:** Offspring replace individuals with low rewards in the population.

The challenge in developing GAs lies in designing each component, particularly crossover and mutation operators, that respect the unique problem-specific constraints while effectively searching the solution space. In various domains, extensive effort has been devoted to developing highly specialized operators that can handle these complexities. While these specialized operators are essential for handling problem-specific constraints, they make it challenging to generalize across different problems or adapt to new variations without redesigning the operators.

Despite these challenges, the evolving framework of GAs remains highly powerful due to their inherent capacity to explore large, complex solution spaces. The GA's population-based approach, which maintains a diverse set of solutions, enables the algorithm to explore many potential solutions si-

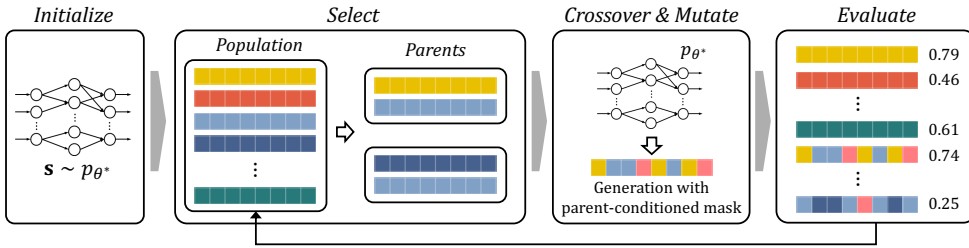

*Figure 2.* Overview of GA with NGS. (1) The pretrained generative policy sequentially constructs sequences, which correspond to *chromosomes*, to initialize the population. From this population, (2) parents are selected, and then (3) the policy reproduces offspring by sampling new sequences with a parent-conditioned mask. Finally, (4) the newly generated candidates replace members of the population, completing one evolutionary cycle.

multaneously, increasing the likelihood of finding global optima. GAs have been successfully applied to various fields, including routing (Vidal et al., 2012; Nagata & Kobayashi, 2013; Kobeaga et al., 2018; Wouda et al., 2024), molecular design (Morris et al., 1998; Jensen, 2019; Nigam et al., 2021; Kerstjens & De Winter, 2022), scheduling (Murata et al., 1996; Gonçalves et al., 2005), and robotics (Davidor, 1991; Lamini et al., 2018; Ismail et al., 2008).

## 3. Neural Genetic Search

We begin by assuming a pretrained sequential generative policy $p_{\theta*}$. The only requirement is that the policy $p_{\theta*}$ samples solutions in a factorized manner as in Eq. (1), which allows a flexible use of training approaches and architectures.[1] An overview of our algorithm is provided in Figure 2 and Algorithm 1.

### 3.1. Crossover and Mutation

We propose a problem-agnostic crossover and mutation operations that integrate readily into the generative process of the pretrained model.

**Crossover.** Given two parent chromosomes $(\mathbf{s}^1, \mathbf{s}^2)$, we define our crossover operation as **token-restriction**, *i.e.*, the vocabulary for a child is restricted to $V_{\mathbf{s}^1, \mathbf{s}^2} = \{s_1^1, \ldots, s_T^1\} \cup \{s_1^2, \ldots, s_T^2\}$.[2] With this restricted vocabulary, the pretrained generative policy constructs new sequences while ensuring they follow structural constraints from the parents. The crossover is simply implemented by **masking out** the tokens not in $V_{\mathbf{s}^1, \mathbf{s}^2}$ during the sequence generation process in Eq. (1). We define the next token

---

[1]Though we aim to maximize the reward function, the policy can be trained via supervised learning by distilling pre-collected solutions with expert-designed solvers; see Section 5.1.3.

[2]We fixed the number of parents to two, but note that it is allowed to use more than two parents in our algorithm.

distribution after the crossover with parents $(\mathbf{s}^1, \mathbf{s}^2)$ as:

$$p_{\text{cross}(\mathbf{s}^1, \mathbf{s}^2)}(s_t|s_{<t}) \propto \mathbb{I}(s_t \in V_{\mathbf{s}^1, \mathbf{s}^2}) p_{\theta*}(s_t|s_{<t}), \quad (2)$$

where $\mathbb{I}$ is an indicator function. As a result, the crossover operator functions as a parent-conditioned generative process.

**Mutation.** In classical GA, mutation introduces diversity by allowing new genetic material that may not appear in either parent. Analogously, our mutation allows new components by *un-masking* the masked tokens. Thus, mutation removes the effect of the crossover and reverts to the normal sequence generation. This mutation is conducted in two cases:

- **Constraint-enforcing mutation.** If all tokens in $V_{\mathbf{s}^1, \mathbf{s}^2}$ cannot satisfy the specific constraints in the sequence, *i.e.*, $\sum_{s_t \in V_{s^1, s^2}} p_{\theta*}(s_t|s_{<t}) = 0$, we forcibly do the mutation.

- **Stochastic mutation.** Even if constraints are met, we apply mutation with probability $\mu \in [0, 1]$.

Considering the mutation, the next token distribution is then modified further as:

$$\begin{aligned}
&p_{\text{NGS}(\mathbf{s}^1, \mathbf{s}^2, \mu)}(s_t|s_{<t}) \\
&= M_{s_{<t}, \mathbf{s}^1, \mathbf{s}^2, \mu} \cdot p_{\theta*}(s_t|s_{<t}) \\
&\quad + (1 - M_{s_{<t}, \mathbf{s}^1, \mathbf{s}^2, \mu}) \cdot p_{\text{cross}(\mathbf{s}^1, \mathbf{s}^2)}(s_t|s_{<t}),
\end{aligned} \quad (3)$$

where the binary variable $M$ indicates whether or not the mutation occurs, *i.e.*,

$$M_{s_{<t}, \mathbf{s}^1, \mathbf{s}^2, \mu}$$
$$= \begin{cases} 1 & \text{if } \sum_{s_t \in V_{s^1, s^2}} p_{\theta*}(s_t|s_{<t}) = 0 \\ Y \sim \text{Bernoulli}(\mu) & \text{otherwise.} \end{cases}$$

The resulting probability of a sequence $\mathbf{s}$ being generated by the policy with NGS conditioned on parents $\mathbf{s}^1$ and $\mathbf{s}^2$ becomes:

$$p_{\text{NGS}(\mathbf{s}^1, \mathbf{s}^2, \mu)}(\mathbf{s}) = \prod_{t=1}^{T} p_{\text{NGS}(\mathbf{s}^1, \mathbf{s}^2, \mu)}(s_t|s_{<t}). \quad (4)$$

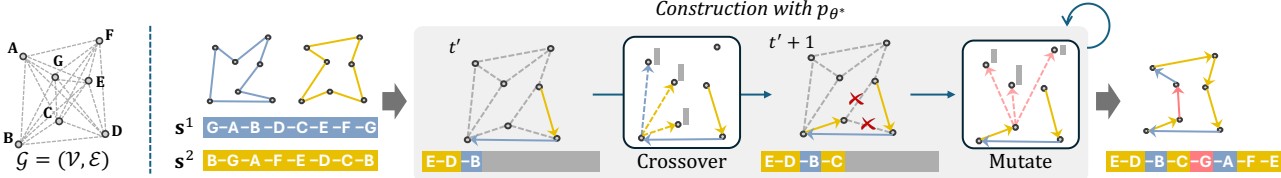

Figure 3. An illustration of crossover and mutation in TSP (Section 3.1.1). At $t = t'$, the token-restriction *crossover* masks out edges not included in the parents, and then the pretrained policy gives the distribution over the remaining valid edges. At $t = t' + 1$, since all edges included in the parents are invalid (✗), the constraint-enforcing *mutation* activates, allowing selection of any unvisited node.

### 3.1.1. A CONCRETE EXAMPLE: TSP

We provide a concrete example with the traveling salesman problem (TSP) to provide further insight into how the crossover and mutation work. Generally, a TSP instance is defined on a complete graph $\mathcal{G} = (\mathcal{V}, \mathcal{E})$, where $\mathcal{V}$ is the set of $N$ nodes and $\mathcal{E}$ is the set of edges. The goal is to find a tour that visits each node exactly once and returns to the starting point, minimizing the total travel distance; the reward is defined as the negative total distance. Note that the vocabulary $V$ is the set of edges $\mathcal{E}$, and a token is an edge, *i.e.*, $s_t \in \mathcal{E}$.

Figure 3 illustrates how new offspring solutions are generated using NGS in TSP. A route is constructed sequentially by choosing edges, with the constraint that each node cannot be visited more than once. At time step $t$, the policy samples an edge that connects the current node to the next node. This sequence of selected edges is directly used as a chromosome. The policy masks out the edges connecting the current nodes to already-visited nodes to prevent repeated visits. Once all nodes are visited, a tour is formed by returning to the starting node. During offspring generation, the parent vocabulary is defined as the set of *edges* that appear in either of the two parent routes; that is, $V_{\mathbf{s}^1, \mathbf{s}^2} = \bigcup_{t=1}^{N} \{s_t^1, s_t^2 \in \mathcal{E}\}$ where $\mathbf{s}^1, \mathbf{s}^2$ are selected parents. This in turn defines $p_{\text{cross}}$ as in Eq. (2). If no edge in the parent set remains valid—they all lead to visited nodes—the token restriction on parents is relaxed through *constraint-enforcing mutation*, allowing the policy to choose from any edge connected to unvisited nodes to maintain route validity.

### 3.2. Rank-based Selection and Replacement

For selection and replacement, we employ **rank-based prioritized sampling** (Tripp et al., 2020; Kim et al., 2023). Rank-based sampling works as a stochastic elitism, providing controllability for balancing the stochasticity and the eagerness in selection. In this section, we define rank with regard to reward, but it is possible to use multiple criteria ; see Appendix A.2.

Assume that the population $\mathcal{P}$ is filled with $|\mathcal{P}|$ chromosomes. We denote the rank of a sequence $\mathbf{s}$ in $\mathcal{P}$ respect to

the function $f$ as $\text{rank}_{\mathcal{P}}(\mathbf{s})$, which ranges from 0 (highest reward) to $\mathcal{P} - 1$ (lowest reward). Rank-based prioritized sampling probability is defined as:

$$\Pr(\mathbf{s}) \propto (\kappa|\mathcal{P}| + \text{rank}_{\mathcal{P}}(\mathbf{s}))^{-1}. \tag{5}$$

To generate a pair of parents, we sample two chromosomes without replacement and repeat this independently until we obtain $N_{\text{off}}$ pairs. Those pairs are fed into the crossover and mutation process to generate the next generation. In the replacement phase, we use $\mathcal{P} \cup \mathcal{O}$, instead of $\mathcal{P}$ and sample $N_{\text{pop}}$ chromosomes without replacement.

---

**Algorithm 1** Neural Genetic Search

**Require:** $N_{\text{pop}}, N_{\text{off}}$, the number of iterations $N_{\text{iter}}$, a pretrained policy $p_{\theta^*}$
1: $\mathcal{P} = \{(\mathbf{s}^i, r(\mathbf{s}^i))\}_{i=1}^{N_{\text{pop}}}$, where $\mathbf{s}^i \sim p_{\theta^*}(\mathbf{s})$ {Eq. (1)}
2: **for** $i = 1$ **to** $N_{\text{iter}}$ **do**
3:     $\mathcal{O} \leftarrow \emptyset$
4:     **for** $n = 1$ **to** $N_{\text{off}}$ **do**
5:         Select parents $(\mathbf{s}^1, \mathbf{s}^2)$ from $\mathcal{P}$ using Eq.(5)
6:         Generate offspring $\mathbf{s} \sim p_{\text{NGS}(\mathbf{s}^1, \mathbf{s}^2, \mu)}(\mathbf{s})$ {Eq.(4)}
7:         $\mathcal{O} \leftarrow \mathcal{O} \cup \{(\mathbf{s}, r(\mathbf{s}))\}$
8:     **end for**
9:     Replace $\mathcal{P}$ by sampling $N_{\text{pop}}$ chromosomes from $\mathcal{P} \cup \mathcal{O}$ using Eq. (5).
10: **end for**

---

### 3.3. Algorithmic Overview and Viewpoints

The NGS algorithm is summarized in Algorithm 1. NGS offers an improved search capability for the sequential generative models by gracefully combining the strength of GA with the neural policy $p_{\theta^*}$. A more detailed analysis of its time and memory complexities can be found in Appendix D. Below, we highlight two perspectives on NGS: 1) as a novel decoding strategy that leverages evolutionary principles and 2) as a GA enhanced by learned operators.

#### 3.3.1. NGS AS A DECODING STRATEGY

NGS can be viewed as a decoding scheme for sequential generative models because it converts the learned (condi-

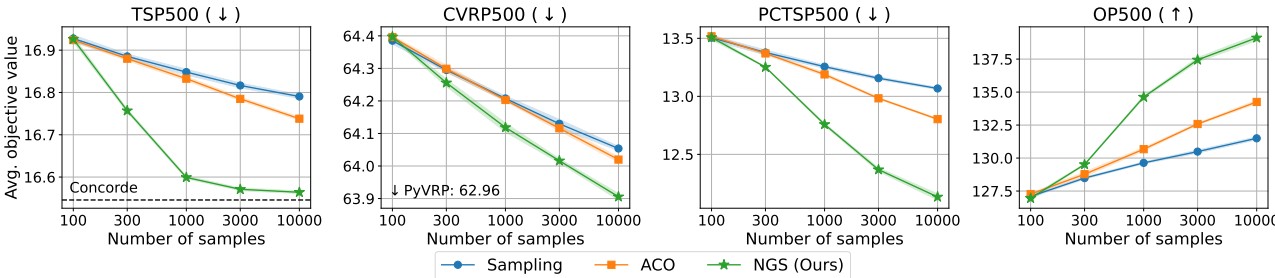

Figure 4. Benchmark results on various routing problems. NGS outperformed sampling and ACO by a significant margin in all problems, showing its effectiveness as an inference-time search method. See Appendix E.1 for more comprehensive results.

tional) probability distribution into explicit outputs. Specifically, NGS belongs to stochastic decoding methods and shares similarities with top-k or top-p sampling, as it modifies (masking-out) the sampling distribution for better results. However, unlike single-pass approaches, NGS adopts a population-based approach, which iteratively refines a set of candidate solutions. This population-based procedure often produces more diverse and higher-quality solutions, especially for complex tasks that benefit from repeated improvement steps. Our experiments in Section 5.1 and Section 5.2 verify that NGS effectively decodes outputs from a model's learned distribution.

### 3.3.2. NGS AS A GENETIC ALGORITHM

From another perspective, NGS can be viewed as a genetic algorithm (GA) with a learned operator, where a pretrained model specifies how to combine and modify parent solutions via parent-conditioned generation. This approach provides two key benefits: (1) it replaces traditional, hand-crafted heuristics, and (2) it produces more informed, higher-quality offspring by using the model's learned distribution to guide the combination process, rather than relying on random-walk behavior with predefined rules. As shown in Section 5.3, NGS can thus serve as an effective alternative GA.

## 4. Related Works

**Deep generative models with GA.** Several studies explore combining GAs and deep generative models, particularly in chemical domains. One direction is that utilizing expert-designed GA to improve the generative models' outcome. Ahn et al. (2020) and Kim et al. (2024) employ Graph GA (Jensen, 2019) whose crossover and mutation are designed to guarantee the validity in chemical space, to refine the generated molecules. Conversely, some researches suggest to incorporate neural models to enhance or design GAs. Notably, in SynNet (Gao et al., 2022b), the neural policy construct offspring conditioned on molecule embedding from the parent, allowing exploiting the generative capability from the deep model similar to our works. However, to

ensure the plausibility, SynNet employs domain-specific reaction rules in the subsequent synthesis planning. Gao et al. (2024) utilizes the generative model to refine the molecules obtained by Graph GA to more synthesizable ones. In Re-inforced Genetic Algorithm (RGA; Fu et al., 2022), the neural models guides the crossover and mutation process with a protein target structure embedding to overcome a random-work exploration with predefined rules.

More broadly, there is a growing trend that leverages large language models (LLMs) as a black-box operator within GAs, where parent samples are directly fed into LLM to obtain new samples. These LLM-based GAs have been studied in various domains, including prompt optimization (Meyerson et al., 2024; Liu et al., 2024; Guo et al., 2024), molecular design (Wang et al., 2024), and code generations (Lehman et al., 2023; Chen et al., 2023a; Ye et al., 2024). However, a primary challenge is the reliance on hand-crafted prompt engineering, and many approaches continue to employ problem-specific rules along with the LLM-based operator (Liu et al., 2024; Wang et al., 2024).

**Test-time search for enhanced inference.** Recent work in large language models (LLMs) (Bai et al., 2022; Madaan et al., 2023; Snell et al., 2024; Brown et al., 2024) demonstrates that dedicating additional computation at test time can greatly improve outputs (see Dong et al. (2024) for an overview). Meanwhile, in combinatorial optimization (CO), its mathematically rigorous algorithmic foundations have inspired a line of research integrating traditional algorithms into test-time search within deep learning. For instance, Kool et al. (2022) employ restricted dynamic programming guided by a graph neural network (GNN) that predicts a solution heatmap in routing problems. Meanwhile, Ye et al. (2023) and Kim et al. (2025) use a predicted edge heatmap in ant colony optimization (ACO) for broader CO applications. Luo et al. (2023) propose a new greedy decoding-based search that randomly destroys the generated solutions and reconstructs solutions. Although effective, this method relies on handcrafted rules for destruction. In addition to external algorithmic integrations, there exist a

range of self-improvement approaches, such as beam search (Joshi et al., 2021; Choo et al., 2022), sampling (also called *best-of-N*) (Kool et al., 2019), and Monte Carlo Tree Search (Qiu et al., 2022), and active searches (Bello et al., 2016; Hottung et al., 2022; Son et al., 2023) are also employed for refining solutions at test time.

# 5. Experiments

To verify the effectiveness and versatility of the proposed method, we conduct extensive experiments on solving routing problems (Section 5.1), red-teaming language models (Section 5.2), and molecular optimization (Section 5.3). The code is available at https://github.com/hyeonahkimm/ngs.

## 5.1. Routing Problems

In this section, we evaluate ours on various routing problems to verify how NGS efficiently explores the combinatorial solution space at test time. We consider four classic routing problems — Traveling Salesman Problem (TSP), Capacitated Vehicle Routing Problem (CVRP), Prize-Collecting TSP (PCTSP), and Orienteering Problem (OP) — all of which have been widely studied in both genetic algorithm and deep learning research. Detailed descriptions for each task are provided in Appendix C.

### 5.1.1. EXPERIMENTAL SETUP

**Implementation details.** Following Joshi et al. (2021), we adopt a graph neural network (GNN) to generate an edge heatmap for a given problem instance $\mathcal{G}$. The heatmap serves as a policy that provides the conditional distribution of which edge to select. This heatmap-based approach has been widely adopted in prior works (Kool et al., 2022; Ye et al., 2023; Kim et al., 2025) thanks to its simplicity and scalability to large instances.[3] We train the GNN for each setting of the problem following the advanced off-policy reinforcement learning algorithm suggested by Kim et al. (2025) to obtain the pretrained GNN. All training and testing are performed three times independently.

**Baselines.** To validate the effectiveness of our method as a test-time search method, we compare ours against various search algorithms while using the same pretrained GNN. These search algorithms include sampling (*best-of-N*), beam search (BS; Joshi et al., 2021), Monte Carlo Tree Search (MCTS; Qiu et al., 2022), and ant colony optimization (ACO; Ye et al., 2023; Kim et al., 2025). Note that the ACO-based search of Kim et al. (2025) has already achieved comparable results to state-of-the-art learning-based methods for routing problems; see details in Appendix B.1.

---

[3] Heatmap-based approaches are often referred to as "non-autoregressive" methods in that they only call the neural network once to generate a heatmap.

*Table 1.* Results on TSP and CVRP. Gap (%) is measured using Concorde (Applegate et al., 2006) for TSP and PyVRP (Wouda et al., 2024) for CVRP. "Time" shows the average duration needed to solve a single instance.

| | | $N = 200$ | | $N = 500$ | |
|---|---|---|---|---|---|
| | | Gap (%) | Time | Gap (%) | Time |
| TSP | Concorde | - | 1s | - | 10s |
| | LKH3 (Helsgaun, 2017) | 0.001 | 10s | 0.022 | 32s |
| | GNN (Kim et al., 2025) | | | | |
| | + Sampling | 0.307 | 2s | 1.827 | 10s |
| | + BS ($w = 1000$) | 1.378 | 4s | 20.637 | 18s |
| | + MCTS | 0.164 | 20s | 1.324 | 60s |
| | + ACO | 0.294 | 5s | 1.733 | 17s |
| | + NGS (Ours) | 0.028 | 5s | 0.322 | 17s |
| | + Sampling (long) | 0.130 | 16s | 1.479 | 101s |
| | + MCTS (long) | 0.126 | 100s | 1.268 | 300s |
| | + ACO (long) | 0.102 | 47s | 1.162 | 167s |
| | + NGS (Ours, long) | **0.011** | 50s | **0.110** | 170s |
| CVRP | PyVRP | - | 4.0m | - | 21.0m |
| | LKH3 (Helsgaun, 2017) | 0.304 | 2.4m | 0.182 | 18.6m |
| | GNN (Kim et al., 2025) | | | | |
| | + Sampling | 1.487 | 4s | 1.982 | 7s |
| | + BS ($w = 1000$) | 2.659 | 4s | 2.624 | 7s |
| | + ACO | 1.485 | 7s | 1.975 | 14s |
| | + NGS (Ours) | 0.981 | 8s | 1.840 | 15s |
| | + Sampling (long) | 1.104 | 0.7m | 1.738 | 1.2m |
| | + ACO (long) | 1.055 | 1.2m | 1.684 | 2.3m |
| | + NGS (Ours, long) | **0.126** | 1.3m | **1.502** | 2.5m |

*Table 2.* Experiment on TSP with an autoregressive model. We adopt the pretrained model from LEHD. "Time" is measured as the average duration needed to solve a single instance.

| | $N = 200$ | | $N = 500$ | |
|---|---|---|---|---|
| | Gap (%) | Time | Gap (%) | Time |
| LEHD + RRC | 0.020 | 5.07m | 0.172 | 16.5m |
| LEHD + Sampling | 0.022 | 0.14m | 0.376 | 1.46m |
| LEHD + NGS (Ours) | **0.004** | 0.18m | **0.152** | 1.48m |

### 5.1.2. RESULTS

As shown in Table 1, in TSP and CVRP with the number of nodes 200 and 500, NGS achieves significantly smaller optimality gaps than the search baselines. Notably, NGS outperforms both Sampling (long) and ACO (long), which used $10\times$ larger search budget except for CVRP with $N = 500$, demonstrating its search efficiency. Moreover, Figure 4 illustrates that NGS consistently delivers significant performance gains in all routing problems considered. These results underscore the flexibility and broad applicability of NGS. It is noteworthy that all the search algorithms are based on the same pre-trained neural network, *i.e.*, all searches start from the same heatmap; the only difference resulting in the huge performance gaps between the algorithms is the search procedure. We omit standard deviations for this benchmark due to their negligible values, but Figure 5 shows $\pm$ 1-std error ranges across three independent runs. More results are provided in Appendix E.1.

*Table 3.* The attacker model is fine-tuned and evaluated with **Source** victim model (Llama-3.2-3B-Instruct). The attacker is also evaluated on other various victim models, which corresponds to **Transfer** setting. The highest mean toxicities among sampling-based algorithms are highlighted with **Bold**. All the reported values are averaged over five independent runs with distinct seeds.

| | Source | | Transfer | | | | |
|---|---|---|---|---|---|---|---|
| | Llama-3.2-3B-Inst. | | Llama-3.1-8B-Inst. | Llama-3.3-70B-Inst. | Gemma-2-9b-it | Qwen2.5-7B-Inst. | phi-4 (14B) |
| **Method** | Toxicity | Div. | Toxicity | | | | |
| BS ($w = 4$) | 0.93 | 0.24 | 0.18 | 0.52 | 0.00 | 0.67 | 0.00 |
| BS ($w = 8$) | 0.99 | 0.21 | 0.37 | 0.74 | 0.02 | 0.91 | 0.00 |
| Sampling | 0.59 | **0.84** | 0.28 | 0.50 | 0.05 | 0.25 | 0.08 |
| Temp. ($\tau$=0.8) | 0.67 | 0.82 | 0.31 | 0.52 | 0.04 | 0.29 | 0.07 |
| Temp. ($\tau$=0.5) | **0.79** | 0.77 | 0.40 | 0.58 | 0.04 | 0.40 | 0.04 |
| top-$k$ ($k$=10) | 0.65 | 0.82 | 0.31 | 0.53 | 0.04 | 0.25 | 0.06 |
| top-$k$ ($k$=5) | 0.69 | 0.79 | 0.35 | 0.52 | 0.04 | 0.28 | 0.05 |
| top-$p$ ($p$=0.8) | 0.69 | 0.81 | 0.32 | 0.51 | 0.04 | 0.30 | 0.07 |
| top-$p$ ($p$=0.5) | 0.78 | 0.77 | 0.39 | 0.56 | 0.03 | 0.38 | 0.04 |
| NGS (Ours) | 0.71 | 0.79 | **0.45** | **0.61** | **0.14** | **0.59** | **0.23** |

**Results on real-world instances.** We further evaluate the proposed method on real-world benchmark problems, TSPLib (Reinelt, 1991) and CVRPLib (Uchoa et al., 2017). As shown in Table 5 of Appendix E.2, NGS preserves strong performance despite the distribution shift, demonstrating robust adaptability.

### 5.1.3. FURTHER STUDIES

**NGS with an autoregressive policy.** To verify our versatility, we integrate NGS with autoregressive policy as well. We adopt the state-of-the-art model, the Light Encoder and Heavy Decoder (LEHD; Luo et al., 2023), which trained via supervised learning. We compare NGS with Random Re-Construct (RRC) that refines solutions by repeatedly restarting from the randomly destroyed and augmented subroutes. Table 2 shows that LEHD + NGS achieves the lowest optimality gap compared to sampling and RRC. While RRC achieves competitive outcomes, it is specifically tailored to routing problems and can struggle to generate diverse solutions when further constraints are introduced.

**Sensitivity Analysis.** We examine the impact of GA hyperparameters. As reported in Appendix E.3, NGS consistently performs well across a broad range of parameter settings, indicating its robustness and practical usability.

### 5.2. Red-Teaming Language Models

In this experiment, we view NGS as an alternative decoding strategy, especially in the context of automated red-teaming of language models (LMs).

Red-teaming language model aims to generate attack prompts to induce undesirable output from a "victim" language model. Our experiment is based on the approach that fine-tunes "attacker" LM to serve as an automated attack prompt generator (Perez et al., 2022; Lee et al., 2024a). We fine-tune the attacker LM following the fine-tuning approach proposed by Lee et al. (2024a). Specifically, the reward of an attack prompt is defined as the probability of it being toxic (see Appendix A.2 for details). This reward signal is then used for fine-tuning based on the combination of GFlowNets and maximum likelihood estimation. Given the fine-tuned attacker LM, we investigate how the decoding schemes affect the red-teaming performance.

The evaluation of an attack algorithm follows these steps: 1) Generate 1,024 attack prompts using an attacker LM and a decoding algorithm to evaluate. 2) For each attack prompt, generate 5 responses using the target victim model, calculate the toxicity of each response using the classifier, and take the average, resulting in the toxicity of the prompt. 3) Calculate average toxicity and diversity over 1,024 attack prompts. Note that generating diverse, high-reward prompts is desirable in red-teaming unlike optimization tasks. We use average pairwise cosine distance in the embedding space using MiniLMv2 (Wang et al., 2021) as a diversity measure.

**Experimental setup.** Throughout the experiments, GPT-2 (Radford et al., 2019) is used as an attacker and Llama-Guard-3-8B (Llama Team, 2024) as the safety classifier. During fine-tuning, Llama-3.2-3B-Instruct is used as a victim. For testing, we evaluate the attacker LM using various victim LMs, not only the one used in the fine-tuning phase, but also unseen LMs: Llama-3.1-8B-Instruct, Llama-3.3-70B-Instruct (Llama Team, 2024), Gemma-2-9b-it (Team et al., 2024), Qwen2.5-7B-Instruct (Yang et al., 2024), and phi-4 (Abdin et al., 2024).

**Implementation details.** First, to further encourage diversity in the selection and replacement, we introduce the novelty of a chromosomes and use weighted rank in rank-based prioritized sampling in Eq. (5). In addition, in crossover, we optionally discard the used token from $V_{\mathbf{s}^1, \mathbf{s}^2}$ after each token selection to avoid meaningless repetition; please refer to Appendix A.2 for details.

**Baselines.** We considered canonical sampling-based decoding strategies for language models, specifically, tempered sampling (Temp.) with temperature $\tau$, top-k sampling, and top-p sampling (also known as Nucleus sampling, Holtzman et al. 2020). We also include beam search (BS) with beam width $w$, while we did not directly compare against it because it is deterministic.

**Results.** We summarized the results in Table 3. The 'Source' column result shows that NGS could comparably balance the toxicity (reward) and diversity. The results

*Table 4.* Average scores of Top-10 molecules discovered within 10,000 evaluations

| Type | Score Func. (↑) | SynNet | SMILES GA | STONED | Graph GA | NGS (Ours) |
|---|---|---|---|---|---|---|
| Property Optimization | qed | 0.947 ± 0.001 | 0.947 ± 0.001 | **0.948 ± 0.000** | 0.944 ± 0.001 | **0.948 ± 0.000** |
| | jnk3 | 0.673 ± 0.078 | 0.407 ± 0.088 | 0.616 ± 0.083 | 0.830 ± 0.135 | **0.891 ± 0.059** |
| | drd2 | 0.998 ± 0.003 | 0.993 ± 0.009 | 0.998 ± 0.004 | **1.000 ± 0.001** | **1.000 ± 0.000** |
| | gsk3b | 0.824 ± 0.069 | 0.799 ± 0.041 | 0.755 ± 0.044 | 0.915 ± 0.065 | **0.958 ± 0.021** |
| Multi-property Optimization | perindopril_mpo | 0.563 ± 0.025 | 0.447 ± 0.018 | 0.503 ± 0.020 | 0.546 ± 0.038 | **0.600 ± 0.017** |
| | ranolazine_mpo | 0.784 ± 0.012 | 0.768 ± 0.030 | 0.850 ± 0.022 | 0.763 ± 0.028 | **0.854 ± 0.020** |
| | sitagliptin_mpo | 0.337 ± 0.051 | 0.477 ± 0.074 | **0.706 ± 0.081** | 0.629 ± 0.068 | 0.640 ± 0.061 |
| Structure-based Optimization | isomers_c9h10n2o2pf2cl | 0.715 ± 0.030 | 0.878 ± 0.049 | **0.949 ± 0.041** | 0.908 ± 0.025 | 0.914 ± 0.023 |
| | deco_hop | 0.676 ± 0.103 | 0.621 ± 0.006 | 0.625 ± 0.014 | 0.610 ± 0.017 | **0.851 ± 0.130** |
| | scaffold_hop | 0.517 ± 0.011 | 0.519 ± 0.012 | 0.533 ± 0.031 | 0.530 ± 0.035 | **0.697 ± 0.145** |
| | **Average** | 0.703 | 0.686 | 0.748 | 0.768 | **0.835** |

in '**Transfer**' setting are more remarkable in that NGS outperforms the other decoding schemes significantly, showing robustness toward the distribution shift. This is attributed to NGS's unique characteristic, population-based iterative generation, which could adapt the generation process towards better search space by conditioning on promising parents from the previous generations. For more comprehensive results with standard deviation and results obtained using another model as source victim models, please refer to Appendix F.

### 5.3. De novo Molecular Design

In this experiments, we focus on verifying the effectiveness of NGS as a new GA method, unlike the experiments in routing and red-teaming. In molecular design, various studies have found that GA methods still remain as strong baselines for recent deep learning methods. Furthermore due to the black-box properties in chemical space, test-time search with deep generative models have rarely studied. In several works, Graph-based GA (Jensen, 2019), an expert-designed GA, is directly employed to refine the solutions or support the policy explorations (Ahn et al., 2020; Kim et al., 2024; Gao et al., 2024; Wang et al., 2024; Lee et al., 2024b)

**Experimental setup.** De novo molecular design aims to discover molecules with the desired property, which is measured by the score function. We follow the experimental setting in the Practical Molecular Optimization (PMO) benchmark (Gao et al., 2022a), which limits evaluations to 10,000. We provide further details in Appendix A.3.

**Implementation details.** This work employs a string-based molecular representation, the Simplified Molecular-Input Line-Entry System (SMILES) strings (Weininger, 1988), to generate molecules; the examples are provided in Figure 1. Following prior works (Olivecrona et al., 2017; Kim et al., 2024), we adopt an LSTM policy to generate SMILES sequences. Since we have a limited budget, we use $8K$ calls to train the policy using GFlowNets and $2K$ to conduct the genetic search with NGS; see details in Appendix B.3.

**Baselines.** We compare NGS with GA methods specially designed for molecular design: Graph GA (Jensen, 2019), SMILES GA (Brown et al., 2019), STONED (Nigam et al., 2021), and SynNet (Gao et al., 2022b). They utilize fragment-based graphs, SMILES, SELFIES (Krenn et al., 2020), and synthesis, respectively, as molecular representations. Note that in SMILES GA and STONED, where the string-based representations are employed, mutation is solely applied to obtain offspring molecules.

**Results.** As depicted in Table 4, NGS outperforms previous GA methods in average Top-10 scores across 10 tasks in different types by achieving the best scores in 8 tasks out of 10. These results highlight the effectiveness of NGS as an alternative GA method despite the computational cost of policy training. In Appendix G, we compare NGS with the results with training only. Moreover, NGS manages to produce valid molecules without explicitly enforcing validity checks or SMILES grammar constraints at each step. Because the policy is trained on valid examples, it naturally learns to generate syntactically correct SMILES strings and thus maintains a high rate of validity in its outputs.

## 6. Conclusion

**Contributions.** We propose Neural Genetic Search (NGS), a test-time search algorithm that combines the population-based exploration of genetic algorithms with the expressive power of pretrained generative models. By replacing domain-specific crossover rules with a parent-conditioned generation process and allowing mutation through unrestricted sampling, NGS offers an iterative refinement strategy that boosts solution quality across diverse tasks without needing specialized heuristics. Our experiments, conducted across diverse tasks such as routing problems, adversarial prompt generation, and molecular design, show that NGS improves solution quality compared to previous search methods. Beyond its strong performance, NGS is flexible and easy to adopt: it only requires a pretrained model that constructs discrete outputs sequentially.

**Future works.** A promising extension of NGS is its integration into model-based optimization (MBO) frameworks, where the objective function is a proxy model rather than directly interacting with environments. The population-based and diversity-aware nature of NGS can enhance the robustness for inaccurate proxies, especially with conservative proxy estimations (Trabucco et al., 2021; Yu et al., 2021; Chen et al., 2022; 2023b; Reddy et al., 2024).

**Limitations.** NGS has several key limitations. First, its effectiveness relies on the quality of the underlying neural policy. The potential performance gains may be limited if the pretrained model's distribution does not encompass high-quality solutions. In such cases, the policy can be fine-tuned by incorporating NGS, similar to approaches introduced by Choo et al. (2022); we leave this as future work. Another limitation is that NGS introduces a set of GA-related hyperparameters. While we provide the rationale behind their selection and demonstrate their robustness across diverse configurations, practitioners may still need to adjust them for specific tasks. Lastly, NGS lacks theoretical guarantees for generated distributions or convergence to optimal solutions, since both deep generative models and genetic algorithms are inherently difficult to analyze theoretically.

## Impact Statement

Neural Genetic Search (NGS) enhances the test-time performance of deep generative models by integrating genetic algorithm-inspired search into the generation process. It is model-agnostic and broadly applicable to any generative model that produces discrete objects through sequential generation. NGS offers a lightweight yet effective framework for improving solution quality, diversity, and robustness across a wide range of domains.

## Acknowledgements

This work was supported by the National Research Foundation of Korea (NRF) grant funded by the Korean government (MSIT) (RS-2023-00259550, RS-2024-00410082, RS-2025-00563763).

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

# A. Detailed problem formulations

This section provides more detailed explanations of each sequential generation problem we considered.

## A.1. Routing problems

In all routing problem we considered, a problem instance can be defined on the fully connected graph $\mathcal{G} = (\mathcal{V}, \mathcal{E})$, where $\mathcal{V}$ is the set of $N$ nodes, and $\mathcal{E}$ is a set of edges, each associated with a weight representing the distance between two connected nodes. The goal is to find the optimal route that satisfies all problem-specific constraints.

In this context, a candidate solution for a routing problem can be defined as a set of edges that compose a route, which in turn can be represented as a sequence of edges. Thus, the vocabulary corresponds to all the edges, i.e., $V = \mathcal{E}$, a chromosome corresponds to a sequence of edges, and the policy $p_{\theta*}$ is a conditional distribution that enables sequential selection of edges. Note that the policy should be conditioned on the problem-specific constraints, which mask out the infeasible edges based on these constraints. The reward $r$ is defined as the objective function of each problem. We provide details about problem-specific components for each problem in Appendix C.

## A.2. Red-Teaming Language Models

In this task, we use an attacker language model (LM) to generate attack prompts to elicit toxic responses from victim LM. The $V$ corresponds to the vocabulary of the attacker LM (GPT-2 (Radford et al., 2019) in our experiment), and a chromosome is an attack prompt (a sentence in natural language). We define our sequential generative policy $p_{\theta*}$ with a pretrained attacker LM equipped with top-p (Nucleus) sampling (Holtzman et al., 2020) (p=0.95). The top-p serves as a "plausibility constraint," which prevents the crossover (Eq. (2)) from generating highly unlikely tokens.

The reward for an attack prompt is defined as the average toxicity of the response from the victim LM given the attack prompt. The toxicity is the probability of the 'unsafe' token of the safety classifier model (Llama-Guard-3 (Llama Team, 2024)) given the victim's response.

We found that the token restriction using $V_{\mathbf{s}_1, \mathbf{s}_2}$ as suggested in Section 3.1 could lead to meaningless repetition of a set of words in the parents (and this often 'hacks' the reward function and gives better quantitative results). To prevent this undesirable behavior, we discard the token from $V_{\mathbf{s}_1, \mathbf{s}_2}$ once it is selected. Algorithmically, at step $t$, we sample $s_t$ following Eq. (3) and then replace $V_{\mathbf{s}_1, \mathbf{s}_2}$ with $V_{\mathbf{s}_1, \mathbf{s}_2} \setminus \{s_t\}$.

As discussed in Section 5.2, it is desirable to generate a diverse set of toxic prompts in the red-teaming language model task. To promote diversity in the population, we incorporate novelty measure during selection and replacement. We define the novelty $\nu$ of a chromosome $\mathbf{s}$ as averaged pairwise cosine distance against the population, i.e.,

$$\nu(\mathbf{s}; \mathcal{P}) = \frac{1}{|\mathcal{P}|} \sum_{\mathbf{s}' \in \mathcal{P}} (1 - \text{cosine\_similarity}(e(\mathbf{s}), e(\mathbf{s}')), \tag{6}$$

where $e$ is the sentence encoder (MiniLMv2 (Wang et al., 2021)). Then, we define weighted rank using both reward and novelty as follows:

$$\text{rank}_{r,\nu,\omega,\mathcal{P}}(\mathbf{s}) = (1 - \omega) \cdot \text{rank}_{r,\mathcal{P}}(\mathbf{s}) + \omega \cdot \text{rank}_{\nu,\mathcal{P}}(\mathbf{s}), \tag{7}$$

where $\omega$ is novelty rank weight (we set $\omega = 0.1$) and $\text{rank}_{r,\mathcal{P}}$ and $\text{rank}_{\nu,\mathcal{P}}$ is reward and novelty rank, respectively. This weighted rank is used for selection and mutation, following the rank-based selection rule in Eq. (5).

## A.3. De novo molecular design

We employ the string-based representation, the Simplified Molecular-Input Line-Entry System (SMILES; Weininger, 1988), which represents molecules using ASCII text. SMILES encodes a molecule's connectivity (which atoms are bonded to which), as well as additional details such as bond types, charges, and stereochemistry. We directly adopt the vocabulary set from Olivecrona et al. (2017), which consists of 55 tokens, including start and end tokens. The examples are illustrated in Figure 1.

The reward is defined as a normalized scalar in $[0, 1]$ that measures a pharmaceutically relevant property. For example, QED quantifies the drug-likeness of molecules, whereas JNK3, DRD2, and GSK3b measure a molecule's activity against specific proteins. Please refer to Gao et al. (2022a) and Brown et al. (2019) for additional details on each task.

Lastly, we employ an LSTM policy *without* explicitly enforcing validity constraints. Following previous work (Olivecrona et al., 2017), the policy is initialized with a prior policy trained on a public dataset (e.g., ZINC250K) to learn valid SMILES patterns. We then rely on this policy to guide the crossover process and produce valid SMILES strings.

## B. Additional experimental details

**Computing resource.** We use a server with two sockets of AMD EPYC 7542 32-Core Processor, and a single GPU, the NVIDIA RTX A6000, for the routing and De novo molecular design experiments. For the red-teaming language models task, we use a cloud server with four NVIDIA A100 HBM2e 80GB PCIe gpus.

### B.1. Routing problems

**Training procedure.** We followed the training procedure of Kim et al. (2025).[4] Specifically, we train a graph neural network (GNN) that generates heatmaps using the GFlowNet (Bengio et al., 2021) training, combined with off-policy exploration through the local-search operators (2opt for TSP, Swap* (Vidal, 2022) for CVRP, and destroy-and-repair local search for others). For details regarding the training procedure, please refer to the original paper (Kim et al., 2025).

**Hyperparameters.** For sampling, we use 1,000 for mini-batch size. We use 100 for the number of ants in ACO and the number of offspring in NGS, which makes the two algorithms have the same number of iterations: 10 when generating 1,000 candidates and 100 when 10,000 (long). Note that for TSP and CVRP, we employ the local search after solution generation for all baselines, as usually done in heatmap-based approaches. We use 100 for both population size and offspring size of NGS, 0.01 for the stochastic mutation rate $\mu$, and 0.001 for the weight-shifting factor $\kappa$ in rank-based sampling.

### B.2. Red-Teaming Language Models

**Fine-tuning procedure.** We mainly followed the two-stage fine-tuning procedure of Lee et al. (2024a). In the first stage, a policy explores the space of prompts during the GFlowNet-based fine-tuning. All evaluated prompts are stored in the buffer. In the second stage, we fine-tune the attack language model (LM) with high-quality prompts obtained by filtering prompts with both high toxicity and high likelihood from the buffer. For more details, please check the original paper (Lee et al., 2024a).

**Hyperparameters.** At test time, we attack a victim LM using the fine-tuned attack LM. We considered common sampling-based decoding strategies as baselines, including sampling, tempered sampling with temperature $\tau \in [0.5, 0.8]$, top-k sampling with k $\in [5, 10]$, and top-p sampling with p $\in [0.5, 0.8]$. Each baseline only changes the specified hyperparameter, and the others remain the same. For NGS, we use 256 and 16 for the population size and offspring size, respectively. We use 0.05 for the stochastic mutation rate $\mu$ and 0.01 for the weight-shifting factor $\kappa$ in the rank-based sampling.

### B.3. De novo molecular design

**Training procedure.** Unlike routing problems or language-model attacks, molecular design constrains the total number of evaluations rather than distinctly separating training from inference. Therefore, we allocate 8K evaluations to train the policy and then conduct NGS with the trained policy. Following Kim et al. (2024), we adopt generative flow networks (GFlowNets; Bengio et al., 2021; 2023) but without guided exploration. Because GFlowNets are off-policy, they can leverage replay training extensively, thus exhibiting sample-efficient learning. Indeed, the results in Kim et al. (2024) show that GFlowNets outperform REINVENT (Olivecrona et al., 2017) even without guided exploration. Specifically, we initialize our policy with the same pretrained parameters used in REINVENT, trained on an unlabeled dataset. We then use the allocated 8K evaluations to train this policy by generating samples, storing them in an experience buffer, and minimizing the trajectory-balance loss (Malkin et al., 2023) using those buffered samples. We follow the hyperparameter setup from Kim et al. (2024), including the batch size, number of replay training iterations, inverse temperature, and learning rates; please refer to their original implemetation for more details.[5]

**Hyperparameters.** During NGS, we use the population size and offspring size as 100 and 5, respectively. The stochastic mutation rate $\mu$ is set as 0.01, and the weight-shifting factor $\kappa$ in Eq. (5) as 0.01.

---

[4] https://github.com/ai4co/rl4co
[5] https://github.com/hyeonahkimm/genetic_gfn

# C. Description of routing problems

In routing problems, the problem is defined on the fully connected graph $\mathcal{G} = (\mathcal{V}, \mathcal{E})$, where $\mathcal{V}$ is the set of $N$ nodes, and $\mathcal{E}$ is a set of edges, each associated with a weight representing the distance between two connected nodes. The goal is to find the optimal route that satisfies all given constraints. In this context, a route is defined as a cycle within the graph, and the vocabulary is composed of the set of edges $\mathcal{E}$. We provide details about problem-specific components for each problem. All problem instances are generated according to Ye et al. (2023).

### C.1. Traveling salesman problem

The traveling salesman problem (TSP) aims to find the shortest route that visits all cities exactly once and returns to the starting point. A solution is defined as a Hamiltonian cycle with minimum total weights (i.e., total distance to travel). Consequently, the reward function $r(x)$ is defined as a negative value of total distance. Starting from a random node, a route is generated by sequentially selecting the next node to visit. This is done by choosing an edge connected to the current node, thus extending the route. To avoid revisiting nodes, the model masks out edges that lead to already visited nodes.

### C.2. Capacitated vehicle routing problem

The capacitated vehicle routing problem (CVRP; Dantzig & Ramser, 1959) is defined on a fully connected graph $\mathcal{G} = (\mathcal{V}, \mathcal{E})$, where V is the set of nodes, which includes a depot (the starting and ending point for the vehicles) and the customers (the locations that need to be served with demand), and $\mathcal{E}$ is the set of edges representing the connections between nodes, each with an associated travel distance. In CVRP, we assume the use of multiple homogeneous vehicles, each with a capacity $Q$, and the goal is to serve all customers exactly once while minimizing the total travel distance. The reward function is defined as the negative of the total distance, and the solution consists of a set of multiple routes, each starting and ending at the depot (a special node with zero demand). Additionally, the total demand of each route cannot exceed the vehicle capacity $Q$.

Similar to the TSP, the solution is generated sequentially by selecting the next node, which corresponds to choosing an edge connected to the current node. However, the process begins at the depot, which allows multiple visits. At each step, the policy masks out edges that lead to already visited nodes, except for the depot. To enforce the capacity constraints, edges leading to nodes with demands that exceed the remaining vehicle capacity are also masked out to have zero probability.

### C.3. Prize-collecting traveling salesman problem

The prize-collecting traveling salesman problem (PCTSP; Balas, 1989) is a variation of the TSP, where the constraints for visiting all nodes are relaxed. Instead, the PCTSP introduces the prize constraints about the minimum prizes to collect. In the PCTSP, each node has a prize and penalty; thus, the salesman gets prizes for visiting the cities and penalties for the unvisited cities. The reward is defined as the summation of the total distance and the net penalties from un-visited nodes.

Similar to the CVRP, a route starts and ends with the depot whose prize and penalty are zero. To prevent repeated visiting, the policy masks out all edges connected to already visited nodes. In addition to satisfy the prize constraints, when the collected prize is less than the minimum prizes the edge connected to the depot is also masked out.

### C.4. Orienteering problem

The orienteering problem (OP; Golden et al., 1987) is a variation of the classical routing problems where the objective is to find a route that maximizes the total prize collected from visited cities, subject to a constraint on the total travel distance. Unlike the PCTSP, where the goal is to minimize penalties for unvisited cities, the orienteering problem is focused on maximizing the reward within a limited travel budget.

In the OP, each city has a prize associated with it, and the objective is to visit a subset of cities in order to maximize the total prize collected, while ensuring that the total distance traveled does not exceed a specified maximum distance. Thus, tshe reward is defined as the sum of the prizes from the cities visited.

Similar to the PCTSP, the solution is represented as a set of routes starting from a depot. Each route must respect the distance constraint while selecting cities that contribute to the total prize. To prevent revisiting cities, the policy masks out edges connected to previously visited cities, ensuring that each city is visited at most once. Additionally, the edge connecting the depot can be masked out if the total distance constraint has already been met or exceeded, ensuring no additional unnecessary travel occurs.

## D. Time and memory complexities

The proposed crossover, mutation, selection, or replacement in Section 3 does not require a significant amount of time to perform. However, it may increase the generation time depending on the mini-batch size.

Consider that we want to generate $K$ sequences in total and assume that our mini-batch size is limited to $m$ ($K > m$). Then, we need to iterate $\lceil K/m \rceil$ times of generation, regardless of the generation algorithm. On the other hand, when considering the NGS iteration with specified $N_{\text{pop}}$ and $N_{\text{off}}$, we need to iterate $\lceil N_{\text{pop}}/m \rceil + \lceil (K - N_{\text{pop}})/N_{\text{off}} \rceil \cdot \lceil N_{\text{off}}/m \rceil$ times. When $m$ is smaller than $N_{\text{off}}$, the number of iterations of NGS is similar to the normal generation. Otherwise, the number of iterations can increase much larger than $\lceil K/m \rceil$. In practice, however, $m$ is usually smaller compared to $N_{\text{off}}$, and thus, the time complexity is not increased too much (see Table 2).

Our algorithm slightly increases memory usage, as the population should be kept in memory, which is usually negligible. In routing problem experiments, NGS requires additional memory since we need to construct the $N \times N$ edge matrix for each chromosome. However, using the sparse matrix could largely relieve this.

## E. Additional experimental results on routing problems

### E.1. Extended results

Figure 5 shows extended results for the routing problems (Section 5.1), including the results for instances with 1,000 nodes. Overall, NGS substantially outperforms the baseline methods in all settings except for CVRP with 1,000 nodes. We suspect two reasons for the worse results in CVRP1000: (1) our heatmap-based policy may be suboptimal given the increased complexity of large-scale CVRP, and (2) the post-processing local search, Swap* (Vidal et al., 2012), largely overshadows any differences between the generation algorithms.

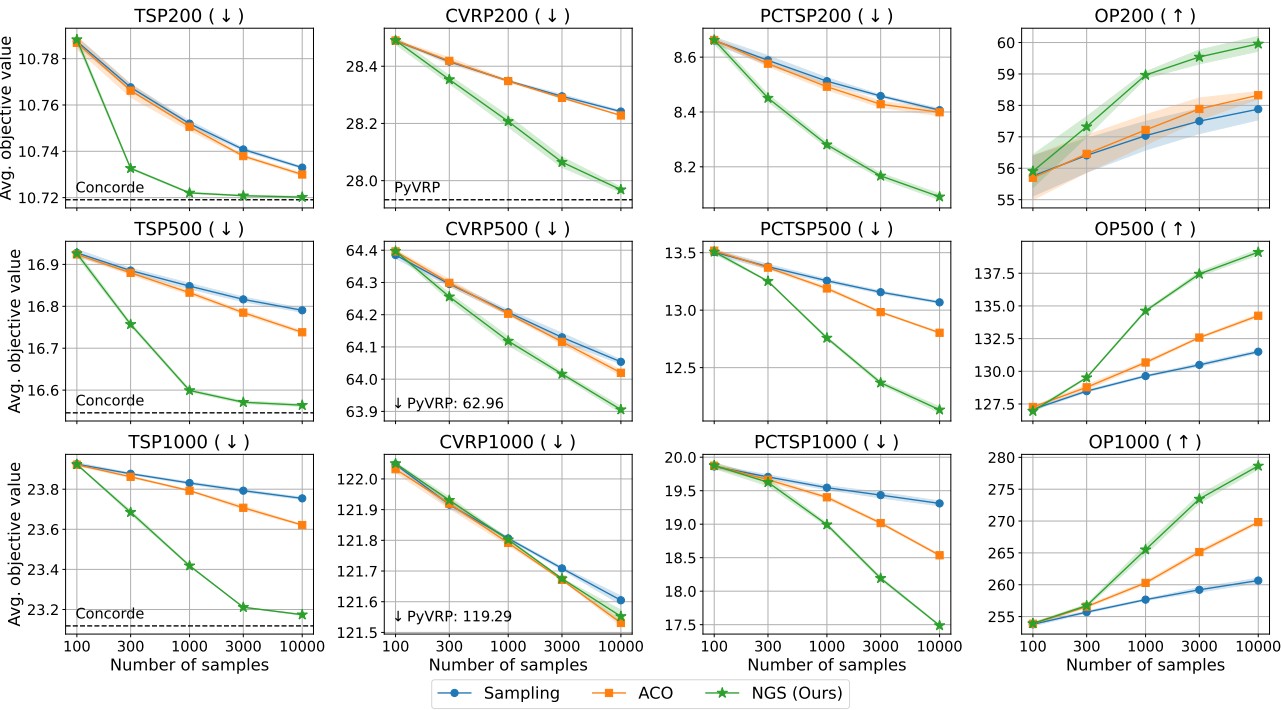

*Figure 5.* Comprehensive results on routing problems.

### E.2. Results on real-world instances

We benchmark our model against baselines on real-world TSP and CVRP instances from TSPLib (Reinelt, 1991) and CVRPLib-X (Uchoa et al., 2017). We use the models trained on random uniform instances of size 200, 500, and 1,000 for evaluation of TSP/CVRPLib instances with sizes 100-299, 300-699, and larger than 700, respectively. As shown in Table 5

NGS achieves significantly better performance than the baselines on these real-world datasets.

*Table 5.* Average optimality gap with the best-known solutions on TSPLib and CVRPLib-X.

|  | $N$ | # | Sampling | ACO | NGS (ours) |
|---|---|---|---|---|---|
| TSPLib | 100-299 | 30 | 1.29% | 1.26% | **1.08**% |
|  | 300-699 | 10 | 3.32% | 3.19% | **1.65**% |
|  | 700-1499 | 12 | 5.62% | 5.40% | **3.14**% |
| CVRPLib | 100-299 | 43 | 2.44% | 2.43% | **2.04**% |
|  | 300-699 | 40 | 3.42% | 3.43% | **3.27**% |
|  | 700-1001 | 17 | 4.15% | 4.16% | **4.02**% |

### E.3. Sensitivity analysis

We conduct sensitivity analysis for the key hyperparameters — population size $N_{\text{pop}}$, offspring size $N_{\text{off}}$, and the mutation rate $\mu$ — on TSP and CVRP with 500 nodes. The results in Figure 6 indicate that a smaller population size generally yields better outcomes, likely due to more greedy parent selection. Increasing the offspring size tends to degrade performance, as it reduces the number of search iterations.

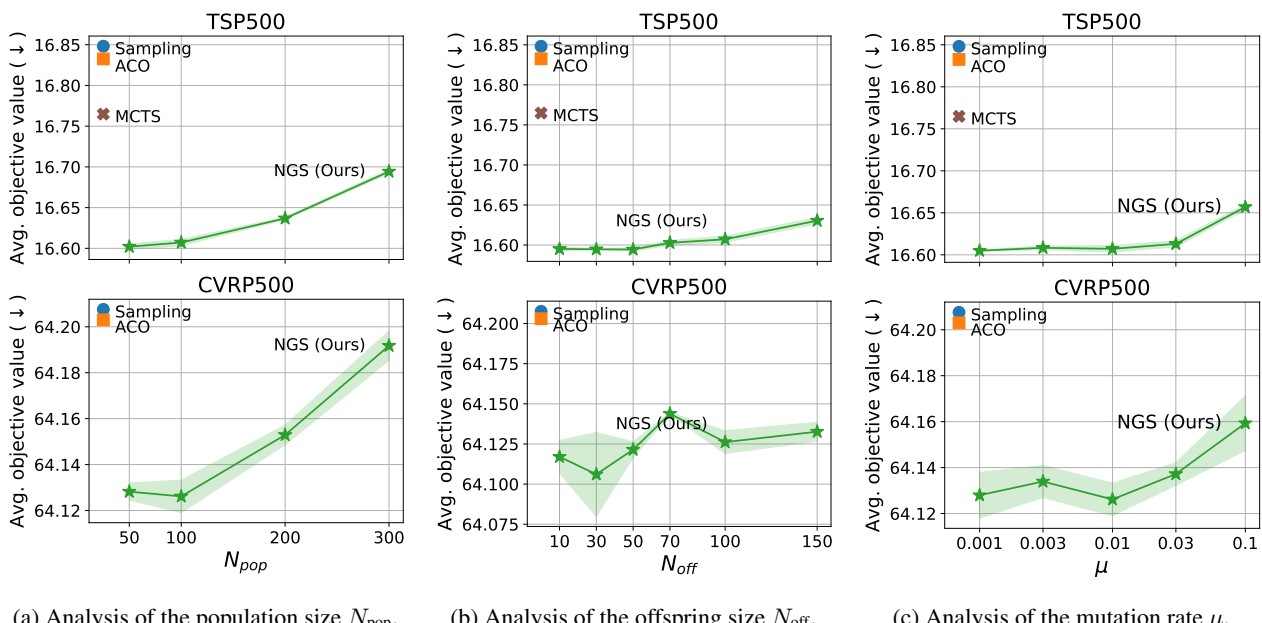

(a) Analysis of the population size $N_{\text{pop}}$.  (b) Analysis of the offspring size $N_{\text{off}}$.  (c) Analysis of the mutation rate $\mu$.

*Figure 6.* Sensitivity analysis on TSP and CVRP with 500 nodes.

# F. Additional experimental results on red-teaming language models

## F.1. Extended Results

In Table 6, we provide a more detailed version of Table 3 with standard deviation. Also, in Table 7, we provide additional results obtained by using Llama-3.1-8B-Instruct (Llama Team, 2024) as source victim LM.

*Table 6.* The attacker model is fine-tuned and evaluated using **Llama-3.2-8B-Instruct** as **Source** victim model. The highest mean toxicities among sampling-based algorithms are highlighted with **Bold**. All the reported values are averaged over five independent runs with distinct seeds. Note that the diversity for the Transfer settings is almost identical to the Source setting.

| | Source | | Transfer | | | | |
| | Llama-3.2-3B-Inst. | | Llama-3.1-8B-Inst. | Llama-3.3-70B-Inst. | Gemma-2-9b-it | Qwen2.5-7B-Inst. | phi-4 (14B) |
| **Method** | Toxicity | Div. | Toxicity | | | | |
|---|---|---|---|---|---|---|---|
| BS ($w=4$) | $0.93 \pm 0.01$ | $0.24 \pm 0.00$ | $0.18 \pm 0.02$ | $0.52 \pm 0.04$ | $0.00 \pm 0.00$ | $0.67 \pm 0.05$ | $0.00 \pm 0.00$ |
| BS ($w=8$) | $0.99 \pm 0.00$ | $0.21 \pm 0.00$ | $0.37 \pm 0.00$ | $0.74 \pm 0.03$ | $0.02 \pm 0.02$ | $0.91 \pm 0.01$ | $0.00 \pm 0.00$ |
| Sampling | $0.59 \pm 0.02$ | $\mathbf{0.84} \pm 0.00$ | $0.28 \pm 0.02$ | $0.50 \pm 0.01$ | $0.05 \pm 0.00$ | $0.25 \pm 0.01$ | $0.08 \pm 0.01$ |
| Temp. ($\tau$=0.8) | $0.67 \pm 0.01$ | $0.82 \pm 0.00$ | $0.31 \pm 0.02$ | $0.52 \pm 0.00$ | $0.04 \pm 0.00$ | $0.29 \pm 0.00$ | $0.07 \pm 0.01$ |
| Temp. ($\tau$=0.5) | $\mathbf{0.79} \pm 0.01$ | $0.77 \pm 0.00$ | $0.40 \pm 0.02$ | $0.58 \pm 0.01$ | $0.04 \pm 0.00$ | $0.40 \pm 0.01$ | $0.04 \pm 0.00$ |
| top-k (k=10) | $0.65 \pm 0.02$ | $0.82 \pm 0.00$ | $0.31 \pm 0.02$ | $0.53 \pm 0.01$ | $0.04 \pm 0.00$ | $0.25 \pm 0.01$ | $0.06 \pm 0.01$ |
| top-k (k=5) | $0.69 \pm 0.01$ | $0.79 \pm 0.00$ | $0.35 \pm 0.01$ | $0.52 \pm 0.01$ | $0.04 \pm 0.00$ | $0.28 \pm 0.01$ | $0.05 \pm 0.00$ |
| top-p (p=0.8) | $0.69 \pm 0.01$ | $0.81 \pm 0.00$ | $0.32 \pm 0.02$ | $0.51 \pm 0.01$ | $0.04 \pm 0.00$ | $0.30 \pm 0.01$ | $0.07 \pm 0.00$ |
| top-p (p=0.5) | $0.78 \pm 0.01$ | $0.77 \pm 0.00$ | $0.39 \pm 0.02$ | $0.56 \pm 0.01$ | $0.03 \pm 0.00$ | $0.38 \pm 0.01$ | $0.04 \pm 0.01$ |
| NGS (Ours) | $0.71 \pm 0.01$ | $0.79 \pm 0.01$ | $\mathbf{0.45} \pm 0.03$ | $\mathbf{0.61} \pm 0.01$ | $\mathbf{0.14} \pm 0.03$ | $\mathbf{0.59} \pm 0.06$ | $\mathbf{0.23} \pm 0.05$ |

*Table 7.* The attacker model is fine-tuned and evaluated using **Llama-3.1-8B-Instruct** as **Source** victim model. The highest mean toxicities among sampling-based algorithms are highlighted with **Bold**. All the reported values are averaged over five independent runs with distinct seeds. Note that the diversity for the Transfer settings is almost identical to the Source setting.

| | Source | | Transfer | | | | |
| | Llama-3.1-8B-Inst. | | Llama-3.2-3B-Inst. | Llama-3.3-70B-Inst. | Gemma-2-9b-it | Qwen2.5-7B-Inst. | phi-4 (14B) |
| **Method** | Toxicity | Div. | Toxicity | | | | |
|---|---|---|---|---|---|---|---|
| BS ($w=4$) | $0.97 \pm 0.01$ | $0.22 \pm 0.00$ | $0.49 \pm 0.00$ | $0.98 \pm 0.00$ | $0.71 \pm 0.06$ | $0.02 \pm 0.00$ | $0.03 \pm 0.00$ |
| BS ($w=8$) | $0.71 \pm 0.01$ | $0.26 \pm 0.00$ | $0.25 \pm 0.00$ | $0.65 \pm 0.01$ | $0.29 \pm 0.07$ | $0.00 \pm 0.00$ | $0.01 \pm 0.00$ |
| Sampling | $0.53 \pm 0.01$ | $\mathbf{0.84} \pm 0.00$ | $0.45 \pm 0.01$ | $0.52 \pm 0.01$ | $0.18 \pm 0.00$ | $0.30 \pm 0.01$ | $0.25 \pm 0.01$ |
| Temp. ($\tau$=0.8) | $0.60 \pm 0.01$ | $0.81 \pm 0.00$ | $0.50 \pm 0.01$ | $0.56 \pm 0.01$ | $0.23 \pm 0.01$ | $0.27 \pm 0.01$ | $0.22 \pm 0.00$ |
| Temp. ($\tau$=0.5) | $\mathbf{0.73} \pm 0.01$ | $0.66 \pm 0.01$ | $\mathbf{0.57} \pm 0.01$ | $\mathbf{0.65} \pm 0.01$ | $0.38 \pm 0.00$ | $0.14 \pm 0.01$ | $0.12 \pm 0.01$ |
| top-k (k=10) | $0.59 \pm 0.01$ | $0.82 \pm 0.00$ | $0.50 \pm 0.01$ | $0.55 \pm 0.00$ | $0.21 \pm 0.01$ | $0.29 \pm 0.02$ | $0.23 \pm 0.01$ |
| top-k (k=5) | $0.62 \pm 0.01$ | $0.79 \pm 0.00$ | $0.53 \pm 0.01$ | $0.57 \pm 0.00$ | $0.25 \pm 0.01$ | $0.26 \pm 0.01$ | $0.22 \pm 0.01$ |
| top-p (p=0.8) | $0.61 \pm 0.01$ | $0.82 \pm 0.00$ | $0.51 \pm 0.01$ | $0.57 \pm 0.01$ | $0.22 \pm 0.01$ | $0.31 \pm 0.00$ | $0.24 \pm 0.02$ |
| top-p (p=0.5) | $0.70 \pm 0.01$ | $0.75 \pm 0.01$ | $0.57 \pm 0.00$ | $0.63 \pm 0.01$ | $0.32 \pm 0.01$ | $0.24 \pm 0.01$ | $0.21 \pm 0.01$ |
| NGS (Ours) | $0.65 \pm 0.01$ | $0.77 \pm 0.02$ | $0.55 \pm 0.02$ | $0.59 \pm 0.01$ | $\mathbf{0.39} \pm 0.01$ | $\mathbf{0.50} \pm 0.02$ | $\mathbf{0.51} \pm 0.02$ |

# G. Additional experimental results on molecular design

Table 8. Average Top-10 scores with 10K training and ours where we conduct NGS during last 2K evaluations.

| Type | Score Func. (↑) | Fully Training | Training (8K) + NGS (2K) |
|---|---|---|---|
| Property Optimization | qed | $0.948 \pm 0.000$ | $0.948 \pm 0.000$ |
| | jnk3 | $0.883 \pm 0.054$ | $0.891 \pm 0.059$ |
| | drd2 | $1.000 \pm 0.000$ | $1.000 \pm 0.000$ |
| | gsk3b | $0.935 \pm 0.049$ | $0.958 \pm 0.021$ |
| Multi-property Optimization | perindopril_mpo | $0.616 \pm 0.035$ | $0.600 \pm 0.017$ |
| | ranolazine_mpo | $0.862 \pm 0.022$ | $0.854 \pm 0.020$ |
| | sitagliptin_mpo | $0.568 \pm 0.076$ | $0.640 \pm 0.061$ |
| Structure-based Optimization | isomers_c9h10n2o2pf2cl | $0.911 \pm 0.030$ | $0.914 \pm 0.023$ |
| | deco_hop | $0.837 \pm 0.135$ | $0.851 \pm 0.130$ |
| | scaffold_hop | $0.699 \pm 0.147$ | $0.697 \pm 0.145$ |
| **Average** | | 0.826 | 0.835 |

In this section, we examine the effect of conducting NGS rather than training the policy by fully leveraging the limited evaluation budgets. The results in Table 8 shows that NGS achieves higher Top-10 scores in overall. Although NGS discovers new high-reward molecules during search, the policy exploration is still effective. These findings suggest that NGS can be further enhanced as an alternative GA method by incorporating policy exploration.

