# OpenReview forum: "Neural Genetic Search in Discrete Spaces"
_ICML.cc/2025/Conference — ICML 2025 poster_

### Official Review · Reviewer_Xt6a · 2025-03-13

**Overall Recommendation:** 2

**Summary:**

The paper proposes a novel test-time search method that integrates genetic algorithm evolutionary mechanisms into deep generative models, particularly for sequential discrete generation tasks. NGS defines crossover as a parent-conditioned generation process, wherein offspring are generated by restricting the vocabulary based on selected parents, and introduces mutation by occasionally removing this restriction to enhance diversity. Experiments are conducted on routing optimization, adversarial prompt generation for language models, and molecular design, and show its effectiveness.

**Claims And Evidence:**

Claims and evidence in the paper look good to me.

**Essential References Not Discussed:**

References look good to me.

**Experimental Designs Or Analyses:**

In results such as Table 1 and Table 2, I would suggest to add a bound/error bar to indicate if the comparisons are significant.

**Methods And Evaluation Criteria:**

1. The problem is defined clearly to me. In section 2.2, I would suggesting to add a flow chart to illustrate GA to audience who is not familiar with it.

2. Line 139-Line 149, why should we put the restriction on vocabulary? For example, in language generation, changing/replacing one single word may make the sentence look better. If we restrict the vocabulary, it seems that some tasks such as text editting can not longer be done.

3. I feel the proposed method in section is exactly the same with general GA, which may indicate that the paper lacks novelty. Also, will the proposed GA converge?

4. In results such as Table 1 and Table 2, I would suggest to add a bound/error bar to indicate if the comparisons are significant.

5. Is there any chance that the algorithm is trapped to a loop?

6. I feel in Table 5, for both CVRP and TSP, the baseline is just Kim et al., 2025, which is too limited as the author mentioned a lot of other works such as Kool et al., 2022; Ye et al., 2023; Kim et al., 2025.

**Other Comments Or Suggestions:**

1. Please add more explanations to the caption of Figure 1. When I read that part, I still got confused about how GA works to generated offsprings.

2. Line 84, right side: "replace" => "replaces".

**Other Strengths And Weaknesses:**

Please refer to my comments in previous sessions.

**Questions For Authors:**

Please refer to my comments in previous sessions.

**Relation To Broader Scientific Literature:**

I feel the paper just proposes a very general genetic algorithm by modifying some of its steps. To me it's definitely related to works that use genetic algorithm for data generation but not that novel.

**Theoretical Claims:**

No heavy theories are involved.

---

> ### Author Rebuttal · Authors · 2025-04-01
>
> ### GA illustration
> > 1. ... add a flow chart to illustrate GA
>
> Thanks for the valuable feedback. We’ve found that Fig 2 can also serve as a flow chart of general GA. We’ll revise the caption to further explain the overall GA process and our distinguished features:
> NGS leverage pretrained generative models within the GA framework. While conventional GAs typically relies on random initialization and problem-specific crossover and mutation, NGS begins with samples from the pretrained models and evolves the population by incorporating novel crossover and mutation mechanisms.
>
> ### Why restrict the vocabulary?
>
> The token-restriction in our **crossover** has the same purpose as crossover in traditional GAs: to focus search on promising regions near high-quality parents. By restricting vocabulary to tokens present in parents, we guide the generation process toward promising regions near parents.
>
> We agree that crossover alone would limit applicability for tasks like text editing. This is why the **mutation** is essential in NGS. When the mutation occurs, token restrictions are removed, allowing the model to introduce tokens not present in either parent. This is how “changing/replacing one single word” happens in our framework.
>
> In practice, we’ve found this balance between restricted crossover and strategic mutation effectively explores the solution space. We’ll make this point clearer in the revised manuscript.
>
> ### Novelty
> > 3. exactly the same with general GA ... lacks novelty
>
> > ... just proposes a very general genetic algorithm
>
> Thanks for raising these important points about novelty.
> First, Genetic Algorithms (GAs) are meta-heuristics that define general search protocols, but their effectiveness depends heavily on how components like crossover and mutation are designed—often requiring significant domain expertise. NGS introduces a novel approach by learning these operators using a sequential generative model, as detailed in Section 3.3.2.
> Second, our work is positioned within the deep learning domain, and our primary contribution is proposing NGS as a test-time search method for sequential generative models. To our knowledge, this is the first approach that integrates GAs directly into the generative process via neural models—unlike prior work that applies GAs to outputs post-generation. A novel combination of a traditional algorithm and a deep generative model should not be underrated.
> We appreciate the reviewer’s recognition of the method’s generality. We will clarify these points and better highlight our contributions in the revised manuscript.
>
> ### Convergence
> > 3. ... will the proposed GA converge?
>
> Unfortunately, we don’t have formal theoretical guarantees for the convergence. Please refer to our response to reviewer r57C (Theoretical guarantee) for detailed discussions regarding this point.
>
> > 5. ... trapped to a loop?
>
> NGS won’t get trapped in a loop as long as the pretrained policy has full support (which is usually true). Our stochastic mutation mechanism ensures that all $x\in\mathcal{X}$ have a positive probability of being generated. Specifically, $p_{\text{NGS}}(x) > 0$ whenever $p_{\theta^*}(x)>0$. This property prevents the algorithm from getting stuck exploring only a subset of the solution space.
>
> ### Bound/Error bar
> > 4. add a bound/error bar
>
> We provided the $\pm$ 1-sigma range over three independent runs in Fig 5. As depicted in the figure, the results are robust to randomness.
>
> We’ll include the comprehensive results with standard deviation for Tables 1, 2, and 5 in our camera-ready version.
>
> ### Baselines in Table 5
> > in Table 5 ... the baseline is just Kim et al., 2025
>
> We limit our baselines to the various search methods that can be used with the **same pretrained model** for fair evaluations. The works you mentioned rely on different pretraining schemes from what we used.
>
> Note that the “ACO” baseline represents [1] ’s performance. NGS significantly outperformed this method, which is notable because [1] already demonstrated superior performance compared to the other works you mentioned. For comprehensive comparisons against other algorithms, please refer to Kim et al. (2025).
>
> [1] Kim et al. “Ant colony sampling with GFlowNets for combinatorial optimization.”, AISTATS, 2025.
>
> ### Suggestions
> > add more explanations to the caption of Figure 1.
>
> Thanks for the suggestion! We will add the following: “Two parent chromosomes (Yellow, Blue) are crossover by following one of the parents at each step of the sequential generation process. Mutation (Pink) occasionally occurs that can take any allowed actions modeled by the generative model.”
>
> Also, please refer to Fig 3 for the detailed generation process. We will also improve the caption of Fig 3.
>
> > "replace" => "replaces".
>
> Thanks. We will fix it.
>
> ---
> **We appreciate your valuable input; please let us know if we can provide any further clarifications.**

---

### Official Review · Reviewer_Synt · 2025-03-13

**Overall Recommendation:** 4

**Summary:**

Neural Genetic Search (NGS) which incorporate ideas from genetic algorithms with generative models as test-time compute method to improve quality/diversity of generation. Crossover: Generating each token autoregressively, masking all tokens not used in the parents. Mutation: random chance of removing the masking token restrictions at each token allowing for more diverse output. It evolves a population akin to GA; given some reward function between 0 and 1, they use rank-based priority sampling to push toward higher rewards. This approach is problem-agnostic, as long as autoregressive token-based models are used and that there is a reward. Its powerful, because it doesn't need backpropagation.

They evaluate NGS on molecule generation, language models, and routing problems comparing to beam search, Monte Carlo Tree
Search, ant colony and other search algorithms.

**Claims And Evidence:**

Claims:
- novel method (yes, as far as I can tell)
- effective test-time search method based on the 3 sets of experiments (yes, they have good evidence)
- NGS can replace conventional decoding with improved robustness (yes, they have good evidence)
- NGS can be viewed as an automated genetic algorithm with learned genetic operators (yes, but they should clarify that this is for autoregressive models with tokens)

**Essential References Not Discussed:**

No

**Experimental Designs Or Analyses:**

See "Methods And Evaluation Criteria"

**Methods And Evaluation Criteria:**

Yes the evaluation and problems tested are very diverse and convincing

**Other Comments Or Suggestions:**

.

**Other Strengths And Weaknesses:**

.

**Questions For Authors:**

.

**Relation To Broader Scientific Literature:**

Fits well with recent trend for test-time compute and its very general and can be applied to various situations, basically any autoregressive model. They properly cite GA with generative models and test-time compute literature as far as I can tell.

**Theoretical Claims:**

There is no strong theory, its a clever algorithm to improve diversity/quality based on a reward without backprop

---

> ### Author Rebuttal · Authors · 2025-04-01
>
> Thank you for your encouraging review. We greatly appreciate your recognition of the novelty and effectiveness of our proposed method, as well as your positive remarks on the diversity and rigor of our experiments.
>
>
> > NGS can be viewed as an automated genetic algorithm with learned genetic operators (yes, but they should clarify that this is for autoregressive models with tokens)
>
>
> We appreciate your suggestion to clarify the scope of our claim regarding NGS as an automated genetic algorithm with learned operators. We will revise the sentence in lines 62-64 (left) to "NGS can be viewed as an automated genetic algorithm that leverages sequential generative models to learn genetic operators, eliminating the need for extensive labor for algorithm design."

---

> > ### Comment · Reviewer_Synt · 2025-04-01
> >
> > Genetic algorithms and evolutionary methods are always perceived and rated negatively at AI conferences because they don't have any theory. But I believe that this work is useful and this one especially is interesting to the community especially with the recent interest in test-time compute. I am writing this just to put things in perspective considering the low-score from the other reviewers. I will keep my score the same.

---

### Official Review · Reviewer_r57C · 2025-03-14

**Overall Recommendation:** 3

**Summary:**

This paper proposes a test-time search method, Neural Genetic Search (NGS), for efficient searching of generative models in discrete spaces. NGS incorporates genetic algorithms into the generation procedure, where the generative model is used to generate offspring conditioned on parents. Experimental results across different domains showcase the effectiveness of NGS.

**Claims And Evidence:**

Most claims in the submission are supported by convincing evidence. However, there are several gaps in the proposed method and experiments to support the novelty and performance of this paper. Please refer to details in "Methods and Evaluation Criteria" and "Experimental Designs or Analyses" sections.

**Essential References Not Discussed:**

- Model-based optimization:

  [1] Designing Cell-Type-Specific Promoter Sequences Using Conservative Model-Based Optimization. NeurIPS 2024.

- Reward-guided sampling:

  [2] Unlocking guidance for discrete state-space diffusion and flow models. ICLR 2025.

  [3] Practical and asymptotically exact conditional sampling in diffusion models. NeurIPS 2023.

  [4] Derivative-free guidance in continuous and discrete diffusion models with soft value-based decoding. arXiv 2024.

**Experimental Designs Or Analyses:**

- In the ablation study in Figure 6, for the mutation rate $\mu$ of NGS, a very small value (0.001) is always preferred. This seems to indicate that the mutation is harmful to the algorithm.
- How does the performance scale with an increasing number of mutations?
- The model lacks comparison with model-based optimization methods and reward-guided methods (see "Methods and Evaluation Criteria").

**Methods And Evaluation Criteria:**

Most parts of the proposed methods and evaluation make some sense. However, there remain several issues:

- The method lacks theoretical guarantees or a rigorous understanding of the generated distribution.
- The optimization target is not clear: is it to generate samples to maximize the evaluation criteria $r(x)$, or to generate reward-tilted distributions $\frac{1}{Z}p_0(x) e^{r(x)/\alpha}$? For each case, the methods to compare, the tasks, and the evaluations are different. For the case of reward maximization, the tasks and evaluations should be robust to reward hacking and there exist model-based optimization methods (eg. [1]) for this. For the case of the reward-guided generative model, the evaluations should consider some measures to verify that the generated sample remains valid samples with respect to the pretrained distribution, and there are many reward-guided sampling methods for this (eg. [2-4]). However, the paper lacks discussion and comparison to these methods.
- An analysis of how the pretrained model helps the parent-conditioned offspring generation and what makes an optimal policy in the generation would be helpful.

[1] Designing Cell-Type-Specific Promoter Sequences Using Conservative Model-Based Optimization. NeurIPS 2024.

[2] Unlocking guidance for discrete state-space diffusion and flow models. ICLR 2025.

[3] Practical and asymptotically exact conditional sampling in diffusion models. NeurIPS 2023.

[4] Derivative-free guidance in continuous and discrete diffusion models with soft value-based decoding. arXiv 2024.

**Other Comments Or Suggestions:**

NA

**Other Strengths And Weaknesses:**

The paper proposes an interesting method that combines evolutionary algorithms and discrete space generative models, and experiments on a variety of domains. I'm open to increasing my score if the author can provide a convincing clarification on the optimization target and a more rigorous understanding of the algorithm, as well as a complete comparison to related baselines.

**Questions For Authors:**

NA

**Relation To Broader Scientific Literature:**

The paper is related to sequential generation, discrete space optimization, and evolutionary algorithms.

**Theoretical Claims:**

There are no theoretical claims in this paper.

---

> ### Author Rebuttal · Authors · 2025-04-01
>
> ### Theoretical guarantees
> > The method lacks theoretical guarantees or a rigorous understanding of the generated distribution.
>
> Thanks for highlighting the importance of theoretical guarantees and analysis.
>
> We acknowledge that rigorous theoretical guarantees, such as formal convergence rates, are indeed valuable. However, deriving such guarantees for deep-learning-based search algorithms is exceptionally challenging. Moreover, in the genetic algorithm literature, existing guarantees are often contingent on idealized assumptions or specific problem classes, making them difficult to generalize widely. We will include this in our limitations of Section 6.
>
> The primary goal of this work is to provide a methodological and empirical contribution. We demonstrate NGS’s flexibility and effectiveness by presenting performance gains on four distinct routing problems, adversarial prompt generation for language models, and molecular optimization tasks. We believe these results clearly illustrate the practical benefits of our proposed approach across various domains.
>
> ### Optimization target and related works
>
> Our goal is to find a discrete object $x$ that maximizes a given objective function $f$ (lines 73-74). More specifically, we aim to find $x^*=\arg\max_{x \in \mathcal{X}} f(x)$, where $\mathcal{X}$ is a set of all possible objects. Our objective function $f$ can be ground truth, noisy simulation, or even proxy models (i.e., model-based optimization).
> - When the objective function is not group-truth-based, we use diversity as additional evaluation criteria; generating more diverse solutions leads to more robustness to the reward hacking (e.g., our red-teaming experiments).
> - NGS can be incorporated into MBO methods that use conservative proxy models, e.g., the one proposed in [1]. Verifying the effectiveness of NGS within the MBO framework is an interesting future work. We will include this discussion in the conclusion. Note that MBO methods are orthogonal to NGS, so they are not included as our baseline.
>
> Thanks for pointing out this important perspective. We will further emphasize our setup in the introduction and at the beginning of Section 3.
>
> [1] Designing Cell-Type-Specific Promoter Sequences Using Conservative Model-Based Optimization. NeurIPS 2024.
>
> ### Analysis on pretrained model
> > An analysis of how the pretrained model helps the parent-conditioned offspring generation and what makes an optimal policy in the generation would be helpful.
>
> To clarify how the pretrained model supports parent-conditioned offspring generation, we refer to Equations (2)–(4). Note that, for simplicity, we omit the explicit notation $\theta^*$ for the pretrained policy.
>
> In Eq. 2, our crossover mechanism works by masking out tokens not present in the parents. So $p_{\text{cross}}$ is essentially the pretrained model with certain tokens masked out based on selected parents $\textbf{s}^1$ and $\textbf{s}^2$. In Eq. 3, we combine $p_{\text{cross}}$ with the original pretrained model $p$ using a binary random variable $M_{s_{t},\textbf{s}^1,\textbf{s}^{2},\mu}$ that indicates whether mutation occurs. If mutation occurs, we sample the next token from the full pretrained model $p$; if not, we sample from the masked version $p_{\text{cross}}$.
>
> Using a pretrained generative model is advantageous:
> - Reward-awareness: Since the model is trained to generate high-reward solutions, it implicitly captures useful patterns for guiding offspring toward promising regions—not just blending parents, but extrapolating toward better candidates.
> - Validity: In contrast to hand-crafted crossover rules, the pretrained model has already learned how to construct valid solutions, so offspring remain feasible without requiring domain-specific heuristics.
>
> As will be discussed in the following question, balancing exploitation (resembling high-reward parents) and exploration (introducing new components via mutation) is important to find optimal solutions.
>
> ### Effect of mutation rate
>
> > ... in Figure 6, for the mutation rate $\mu$ of NGS, a very small value (0.001) is always preferred.
> > How does the performance scale with an increasing number of mutations?
>
> Mutation encourages explorations during the generation procedure by removing the token restriction from crossover. Using a too-low value of $\mu$ makes it hard to escape from local optima.
>
> Although the results in Fig 6c for TSP seem to give better results with a lower mutation rate, this trend is not consistently observed in CVRP or the red-teaming LM task.
>
> We have included additional experiments analyzing the impact of different mutation rates (𝜇) on the red-teaming LM task using Llama-3.2 victim. As shown in the table below, our method remains robust across a range of 𝜇 values.
>
> |$\mu$|Toxicity|Diversity|
> |-|-|-|
> |0.01|0.699 ± 0.029|0.790 ± 0.013|
> |0.02|0.698 ± 0.009|0.791 ± 0.011|
> |0.05|**0.712** ± 0.008|0.791 ± 0.015|
> |0.1|0.681 ± 0.015|0.801 ± 0.010|
> |0.2|0.659 ± 0.012|**0.804** ± 0.008|

---

> > ### Comment · Reviewer_r57C · 2025-04-05
> >
> > Thank the authors for the rebuttal. I increased my score.

---

### Decision · Program_Chairs · 2025-05-01

**Decision:**

Accept (poster)

**Comment:**

This paper incorporates methods from genetic algorithms into the (test time) generative process of autoregressive models. Notably, they define a "crossover" operation which limits the vocabulary of children conditioned on that of the parents. The method is a nice contribution in the sense that it bring ideas from GA into generative modeling in language which expand on the functionality of existing models.

There are concerns about theoretical guarantees, but I think the authors are right that lacking these guarantees should not signify that the main ideas here have value: both in the context of deep learning and GA said guarantees (particularly with LLMs) are hard to quantify. Other than that, I think that there's potential for future contributions as tokenized autoregressive models are only becoming more popular (such as in vision and RL).